

# Prognostic implications of metabolism-associated gene signatures in colorectal cancer

Yandong Miao[1,*], Qiutian Li[2,*], Jiangtao Wang[1], Wuxia Quan[3], Chen Li[4], Yuan Yang[1] and Denghai Mi[1,5]

[1] The First Clinical Medical College, Lanzhou University, Lanzhou City, Gansu Province, PR China
[2] Department of Oncology, The 920th Hospital of the Chinese People's Liberation Army Joint Logistic Support Force, Kunming City, Yunnan Province, PR China
[3] Qingyang People's Hospital, Qingyang City, Gansu Province, PR China
[4] The 3rd Affiliated Hospital, Kunming Medical College, Tumor Hospital of Yunnan Province, Kunming City, Yunnan Province, PR China
[5] Gansu Academy of Traditional Chinese Medicine, Lanzhou City, Gansu Province, PR China
[*] These authors contributed equally to this work.

## ABSTRACT

Colorectal cancer (CRC) is one of the most common and deadly malignancies. Novel biomarkers for the diagnosis and prognosis of this disease must be identified. Besides, metabolism plays an essential role in the occurrence and development of CRC. This article aims to identify some critical prognosis-related metabolic genes (PRMGs) and construct a prognosis model of CRC patients for clinical use. We obtained the expression profiles of CRC from The Cancer Genome Atlas database (TCGA), then identified differentially expressed PRMGs by R and Perl software. Hub genes were filtered out by univariate Cox analysis and least absolute shrinkage and selection operator Cox analysis. We used functional enrichment analysis methods, such as Gene Ontology, Kyoto Encyclopedia of Genes and Genomes, and Gene Set Enrichment Analysis, to identify involved signaling pathways of PRMGs. The nomogram predicted overall survival (OS). Calibration traces were used to evaluate the consistency between the actual and the predicted survival rate. Finally, a prognostic model was constructed based on six metabolic genes (NAT2, XDH, GPX3, AKR1C4, SPHK1, and ADCY5), and the risk score was an independent prognostic prognosticator. Genetic expression and risk score were significantly correlated with clinicopathologic characteristics of CRC. A nomogram based on the clinicopathological feature of CRC and risk score accurately predicted the OS of individual CRC cancer patients. We also validated the results in the independent colorectal cancer cohorts GSE39582 and GSE87211. Our study demonstrates that the risk score is an independent prognostic biomarker and is closely correlated with the malignant clinicopathological characteristics of CRC patients. We also determined some metabolic genes associated with the survival and clinical stage of CRC as potential biomarkers for CRC diagnosis and treatment.

Corresponding author
Denghai Mi, mi.dh@outlook.com

## INTRODUCTION

Cancer is associated with multiple genes, and the accumulation of molecular modifications in the genome of somatic cells is the basis of cancer progression (*Vogelstein & Kinzler, 2004*). Cancers of all types are the primary cause of death globally and one of the most significant obstacles to increased life expectancy (*World Health Organization, 2018*). Colorectal cancer (CRC) is the fourth most fatal carcinoma, accounting for almost 900,000 deaths annually (*Dekker et al., 2019*). It is the second most frequently diagnosed cancer in women and the third most frequently diagnosed in men (*Christopher, 2018*). The development of CRC is a complicated biological process requiring a constellation of factors, including lifestyle, obesity, and environmental influences that may be associated with CRC, and which involves profound shifts at various molecular levels, including in the genome, transcriptome, methylation, and epigenome. The application of second-generation DNA sequencing techniques through whole-genome, whole-exome, and whole-transcriptome approaches leads to significant advances in cancer genomics (*Meyerson, Gabriel & Getz, 2010*). The Cancer Genome Atlas (TCGA) has created an unprecedented opportunity to investigate cancer biology with clinical significance (*Liu et al., 2018*).

Cellular metabolism is the foundation of all biological activities and participates in the regulation of cell growth and proliferation (*Zhu & Thompson, 2019*). Cancer cells reprogram their metabolism to utilize nitrogen and carbon to synthesize the macromolecules necessary for the growth and proliferation of tumors (*Keshet et al., 2018*). As mentioned before, CRC is a multi-step development in which several genetic events drive the initiation and progression of CRC. Specific metabolic pathways can also affect the occurrence and progression of CRC (*La Vecchia & Sebastián, 2020*). Remarkably, some genetic drivers of CRC are well-known regulators of cancer metabolism, such as p53 (*Labuschagne, Zani & Vousden, 2018*), KRas (*Kawada, Toda & Sakai, 2017*), and Wnt (*Thompson, 2014*). A recent report also manifested that knockdown of MYC in CRC cells can reset the altered metabolism and suppressed cell growth (*Satoh et al., 2017*). Oncogene signals drive the progression of CRC associated with the control of specific metabolic pathways in CRC (*Hutton et al., 2016*; *Pate et al., 2014*; *Satoh et al., 2017*). Metastatic CRC cells participate in a selective metabolic adaptation to efficiently form liver metastasis (*Loo et al., 2015*). Metabolic genes also play essential role in the epithelial-mesenchymal transition (*Shaul et al., 2014*). Previous research in metabolic genes related to CRC has garnered significant attention, and metabolic gene variants promote colorectal carcinogenesis (*Hlavata et al., 2010*; *Hong et al., 2018*; *Pommier et al., 2016*). Transcription and genomic data from many tumor samples have been used to research tumor metabolism and explore the role of genes to promote metabolic reprogramming in tumors (*Arif et al., 2017*; *Dejure & Eilers, 2017*; *Ying et al., 2012*). However, few studies have focused on constructing a prognostic model based on differential metabolic genes combined with the prognostic prediction model in CRC.

We attempted to construct a prognostic model of prognosis-related metabolic genes (PRMGs) and explore the prognostic value of these PRMGs in CRC. The colorectal cancer-specific prognostic model was used to determine some pivotal PRMGs for the
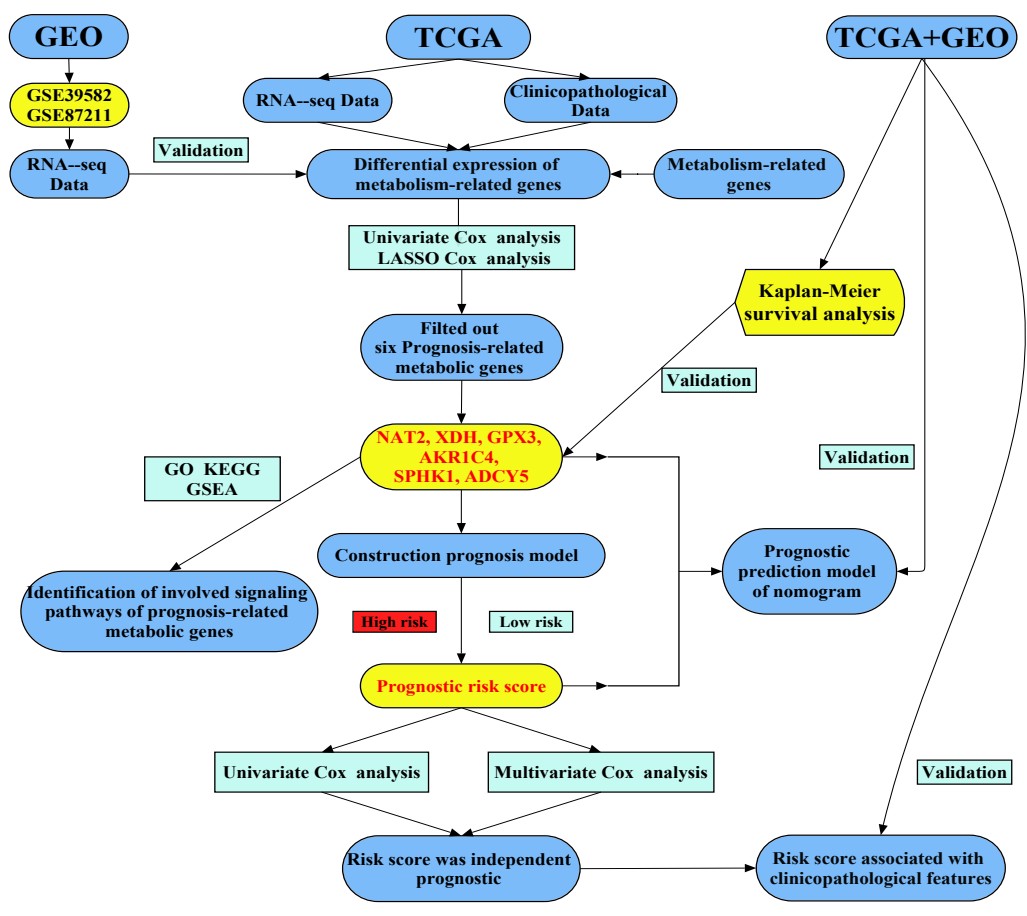

**Figure 1** A flow chart of the study design and analysis.

diagnosis and treatment of CRC and to identify novel potential targets. We also validated the PRMG signature in an independent colorectal cancer cohort. We identified a new essential marker (risk score) to predict the prognosis of CRC patients. Our study developed a new risk model as an independent prognostic biomarker in risk stratification for CRC patients. The flow chart of the study design and analysis is shown in Fig. 1.

# MATERIALS AND METHODS

## Data sources

Both the RNA-sequencing datasets and clinical data of CRC were obtained from the TCGA database (Data Release 22.0—January 16, 2020, https://tcga- data.nci.nih.gov/tcga/). The two main filter criteria for our data were as follows: (1) The keywords of cases are "colon, rectosigmoid junction, and rectum [Primary Site]", "TCGA[Program]", "TCGA-COAD, TCGA-READ [Project]", "Adenomas and Adenocarcinomas [Disease Type]".(2) The keywords of files are "Transcriptome Profiling [Data Category]", "Gene Expression Quantification [Data Category]", "RNA-Seq [Experimental Strategy]", "HTSeq—FPKM [Workflow Type]", "Clinical [Data Category]", "BCR XML [Data Format]". The matrix files of RNA-sequencing for different samples were collated and annotated onto the

genome. The expression of mRNA was extracted from the matrix file obtained from the RNA-sequencing data. The data access used GSE39582 and GSE87211 as the validation data obtained from the Gene Expression Omnibus (GEO, https://www.ncbi.nlm.nih.gov/geo/). Raw data of gene chips are normalized using the RMA algorithm provided by "limma" (*Ritchie et al., 2015*). Perl and R-package "sva" were used to merge microarray data and reduce heterogeneity between the two studies. Metabolism-related genes were obtained from KEGG symbols of the Gene Set Enrichment Analysis (GSEA) website (https://www.gsea-msigdb.org). R software (version 3.6.2) was used for data annotation and the extraction of metabolic gene expression for the TCGA and GEO data.

## Identification of prognosis-associated metabolism-related genes

Data extraction and integration were performed using Perl software. We screened differentially expressed metabolism-related genes using the Wilcox Test with R-package "edgeR" (*Robinson, McCarthy & Smyth, 2010*), and "limma". |Log Fold Change| > 1.5 and False Discovery Rate (FDR) < 0.05 were set as the cutoff. Bidirectional hierarchical clustering was analyzed and we drew a heatmap using R-package "pheatmap" (https://cran.r-project.org/web/packages/pheatmap/). R-package "ggplot2" was used to draw a volcano map. Prognosis-associated metabolism-related genes were identified by univariate Cox regression analysis.

## Construction prognosis model of metabolism-related genes and survival analysis

We screened out the metabolism-related genes that had a significant correlation with overall survival (OS) of colorectal cancer cohorts using univariate Cox regression analysis ($p < 0.01$). The least absolute shrinkage and selection operator (Lasso) Cox regression analysis constructed the optimal model of metabolism-related genes using the R-package "glmnet" (*Friedman, Hastie & Tibshirani, 2010*). Genes expression and survival analysis were evaluated using the Kaplan–Meier method and the log-rank test ($p < 0.05$). The risk score for each patient calculated as: Risk score $= \sum_{j=1}^{n} Coef_j * X_j$, $Coef_j$ denotes the coefficient and $X_j$ denotes the expression levels of each metabolism-related gene (*Liu et al., 2019*). Survival data were selected from each sample obtained from the clinical data downloaded from the TCGA and combined with the previously acquired expression profiling data. The median risk score was selected as a cutoff value to create colorectal cancer cohorts. The survival curve was drawn according to the high and low-risk value by R-package "survival" "survminer". The R-package "survival ROC" drew the Receiver Operating Characteristic (ROC) curve, which was used to investigate the sensitivity and specificity of the survival prediction by the gene marker risk score (*Le, Yapp & Yeh, 2019*). Area Under Curve (AUC) served as an indicator of prognostic veracity (*Le, 2019*; *Sachs, 2017*). The risk curve, survival state diagram, and heatmap were drawn based on the different risk scores of the patients. Independent prognostic metabolic genes were recognized using univariate and multivariate Cox proportional risk regression analysis. We conducted clinical correlation an analysis using the R-package "beeswarm". The prognostic metabolism-related gene was verified in the independent colorectal cancer cohort of GEO (GSE39582, GSE87211) (*Hu et al., 2018*; *Marisa et al., 2013*).

## Functional enrichment analysis of metabolism-related genes

Functional enrichment analysis of metabolic genes was conducted based on the DAVID database (https://david.ncifcrf.gov/home.jsp) (*Huang, Sherman & Lempicki, 2009*), which identified Gene Ontology (GO) categories in the biological processes (BP), cellular components (CC), or molecular functions (MF). The DAVID database was also used to determine the Kyoto Encyclopedia of Genes and Genomes (KEGG) pathways. FDR < 0.05 was the cutoff value. R-package "GOplot" and "ggplot2" were used to integrate expression data and functional and pathway enrichment analysis. Gene Set Enrichment Analysis (GSEA) was also used to uncover the signaling pathways and biological processes of differentially expressed genes between the high and low-risk subgroups. The number of permutations was set to 1,000, and FDR < 0.25 was considered to be statistically significant.

## Exploitation of the nomogram

Age, gender, stage, TNM stage, and risk score were used to draw a nomogram using the R-packages "Hmisc", "lattice", "Formula", "ggplot2", "foreign" and "rms". Calibration traces were used to assess the consistency between the actual and predicted survival rates. The Consistency index (C-index), ranging from 0.5 to 1.0, was calculated to evaluate the model's capability for predicting an accurate prognosis. Measurements of 0.5 and 1.0 from the model represent a random probability or an excellent performance for predicting survival, respectively.

Statistical analysis was performed using the R software (Version 3.6.2; https://www.R-project.org). *P*-values for all of the analyses were: less than 0.05 (statistically significant), 0.01 (more statistically significant), and 0.001 (most statistically significant).

# RESULTS

## Characteristics of patients

The TCGA CRC cohort consisted of 624 patients. Patients who have not survival time and TNM stage were excluded, resulting in a total of 596 patients. Patients were further screened, and those whose survival time was less than 90 days were also excluded, resulting in a total of 545 patients. A total of 692 mRNA expression profiles of CRC were downloaded from TCGA. Among them, 51 (7.4%) derive from healthy samples, and 641 (92.6%) come from tumor samples.

The Marisa cohort consisted of 585 patients (GSE39582), including 566 with stage I-IV colon adenocarcinoma and 19 colon mucosa samples (*Marisa et al., 2013*). The Hu cohort included 363 patients (GSE87211), consisting of 203 rectal tumors and 160 rectal mucosa samples (*Hu et al., 2018*). A total of 948 patients were obtained after merging the two sets of data. Patients with incomplete information were excluded, resulting in a total of 720 tumor patients. Patients were further screened and those whose survival time was less than 90 days were also excluded, resulting in a total of 701 tumor patients. The demographic and clinical characteristics of patients are listed in Table S1.

## Construction prognosis model for TCGA colorectal cancer cohort

A total of 102 differentially expressed metabolism-associated genes were screened containing 40 upregulated and 62 down-regulated genes. Gene expression profiles of

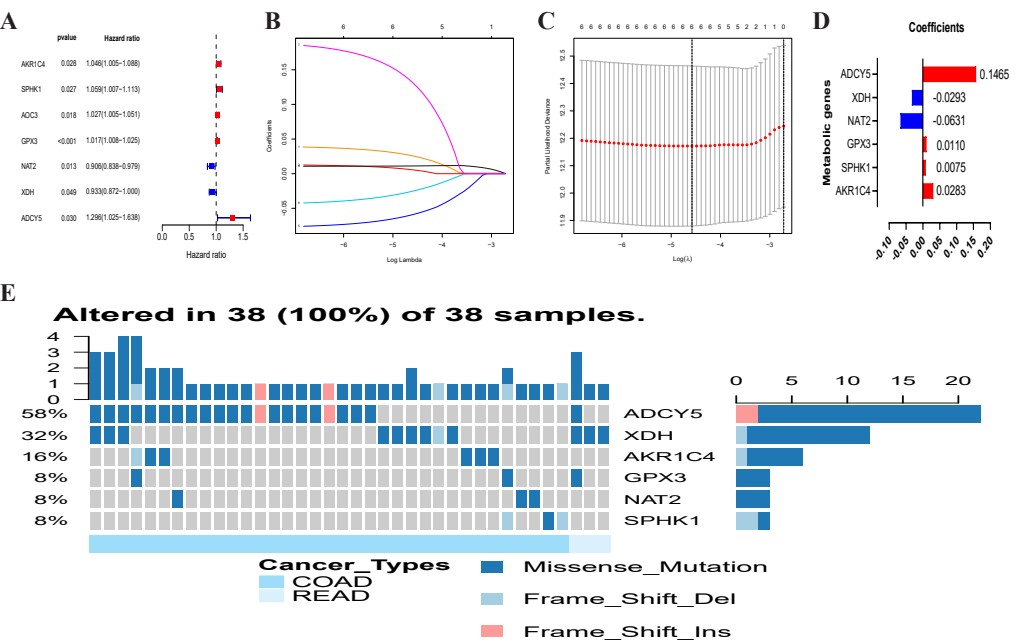

**Figure 2** **Establishment of prognostic metabolic gene signature by univariate and LASSO Cox regression analysis.** (A–D) The process of constructing the signature containing six metabolic genes. (A) The HR, 95% CI calculated by univariate Cox regression. A coefficient (D) profile plot was generated against the log (lambda) sequence (B). (C) Selection of the optimal parameter (lambda) in the LASSO model for colorectal cancer. (E) Genetic alteration of the six genes in the colorectal cancer cohort. *X* axis represents cancer type, sky blue indicates COAD, light blue indicates READ. The left *Y* axis represents ratio of gene mutation, right *Y* axis represents gene names. Dark blue, cyan, and pink small rectangles indicate the type of gene mutation. HR, hazard ratios; CI, confidence intervals; COAD, colon adenocarcinoma; READ, rectum adenocarcinoma.

102 genes were displayed separately for colorectal cancer cohorts in normal and tumor tissues in Fig. S1. Univariate Cox regression analysis revealed seven metabolism-related genes that were significantly related to the OS of CRC. NAT2 and XDH were low-risk PRMGs (Hazard Ratio (HR) < 1, $p < 0.05$); GPX3, AOC3, AKR1C4, SPHK1, and ADCY5 were high-risk PRMGs (HR > 1, $p < 0.05$) (Fig. 2A).

Six PRMGs were filtered out by LASSO COX regression analysis and the coefficient was calculated. The model provided the best figure when six genes were included (Figs. 2B and 2C). The coefficient of each CRC gene is shown in Fig. 2D. The prognosis model was constructed based on these six genes (NAT2, XDH, GPX3, AKR1C4, SPHK1, ADCY5). The full names, pathways, and coefficients of these genes are shown in Table 1.

We investigated genetic alterations in the role of CRC risk-related metabolic genes in CRC using Gene Set Cancer Analysis (GSCA) Lite (http://bioinfo.life.hust.edu.cn/web/GSCALite/). The genes of interest in CRC were changed in 38 of 38 queried samples (100%) (Fig. 2E). The frequent genetic changes indicated the crucial roles of these genes in the development of CRC.

Each patient's risk score was calculated based on the mRNA expression level and risk coefficient of each gene. The calculation of the risk score was: risk score = 0.0283 ×

**Table 1  Functions of genes in the prognostic gene signatures.**

| Gene symbol | Full name | Pathway | Risk coefficient |
|---|---|---|---|
| AKR1C4 | Aldo-keto reductase family 1 member C4 | Participate in multiple hormone metabolism processes | 0.0283 |
| SPHK1 | Sphingosine kinase 1 | Sphingolipid metabolism | 0.0075 |
| GPX3 | Glutathione peroxidase 3 | Glutathione metabolism | 0.0110 |
| NAT2 | N-acetyltransferase 2 | Drug metabolism—cytochrome P450 | −0.0631 |
| XDH | Xanthine dehydrogenase | Involved in the oxidative metabolism of purines | −0.0293 |
| ADCY5 | Adenylate cyclase 5 | Links between intrauterine growth and adult height and metabolism | 0.1465 |

expression of AKR1C4 + 0.0075 × expression of SPHK1 + 0.0110 × expression of GPX3 + (−0.0631) × expression of NAT2 + (−0.0293) × expression of XDH + 0.1465 × expression of ADCY5. The risk score was used to predict prognosis. Patients were divided into a high-risk group and a low-risk group with the median risk score as the cut-off value. A heatmap was developed to show the gene expression profiles of the high- and low-risk TCGA-CRC groups (Fig. 3A) and the GEO-CRC group (Fig. 3B). The risk score had a significant correlation with age, T, N, M, and clinical stage in TCGA-CRC, and age, gender, T, N, and clinical stage in GEO-CRC. The high-risk genes (GPX3, AKR1C4, SPHK1, ADCY5) were more likely to be expressed in the high-risk group patients, while low-risk genes (NAT2 and XDH) were expressed in the low-risk group patients of TCGA-CRC (Fig. 3A) and GEO-CRC (Fig. 3B). The association between survival time and risk score is shown in Fig. 3C (TCGA-CRC), 3D (GEO-CRC). The survival time decreased as risk score increased and the higher the risk score, the more deaths in TCGA-CRC (Fig. 3E) and GEO-CRC (Fig. 3F). Risk scores both were significantly related to OS in the univariate independent prognostic analysis of TCGA-CRC (HR = 3.324, 95% CI [1.956–5.650], $P < 0.001$) (Fig. 3G) and GEO-CRC (HR = 2.096, 95% CI [1.383–3.177], $P < 0.001$) (Fig. 3H).

The multivariate independent prognostic analysis showed that the risk score was an independent prognostic predictor in the TCGA-CRC group (HR = 2.639, 95% CI [1.413–4.928], $P = 0.002$) (Fig. 4A) and the GEO-CRC group (HR = 1.658, 95% CI [1.059–2.598], $P = 0.027$) (Fig. 4B). Kaplan–Meier cumulative curves indicated that the OS rate between the high-risk group and the low-risk group was significantly different. The survival time of patients with high-risk score was significantly shorter than that of patients with low-risk score in the TCGA-CRC group ($P < 0.001$) (Fig. 4C) and GEO-CRC group ($P < 0.001$) (Fig. 4D). The 1-, 3-, and 5-year AUC values of the risk score were 0.672, 0.608, and 0.648, respectively, and the prognostic accuracy of the stage was higher than other clinical characteristics in TCGA-CRC (Figs. 4E, 4G and 4I). Similarly, the 1-, 3-, and 5-year AUC values of the risk score were 0.612, 0.566, and 0.573, respectively, and the prognostic accuracy of the M-stage was higher than other clinical characteristics in GEO-CRC (Figs. 4F, 4H and 4J). These results confirm that six metabolic gene signatures can also predict survival in the independently validated GEO colorectal cohort.

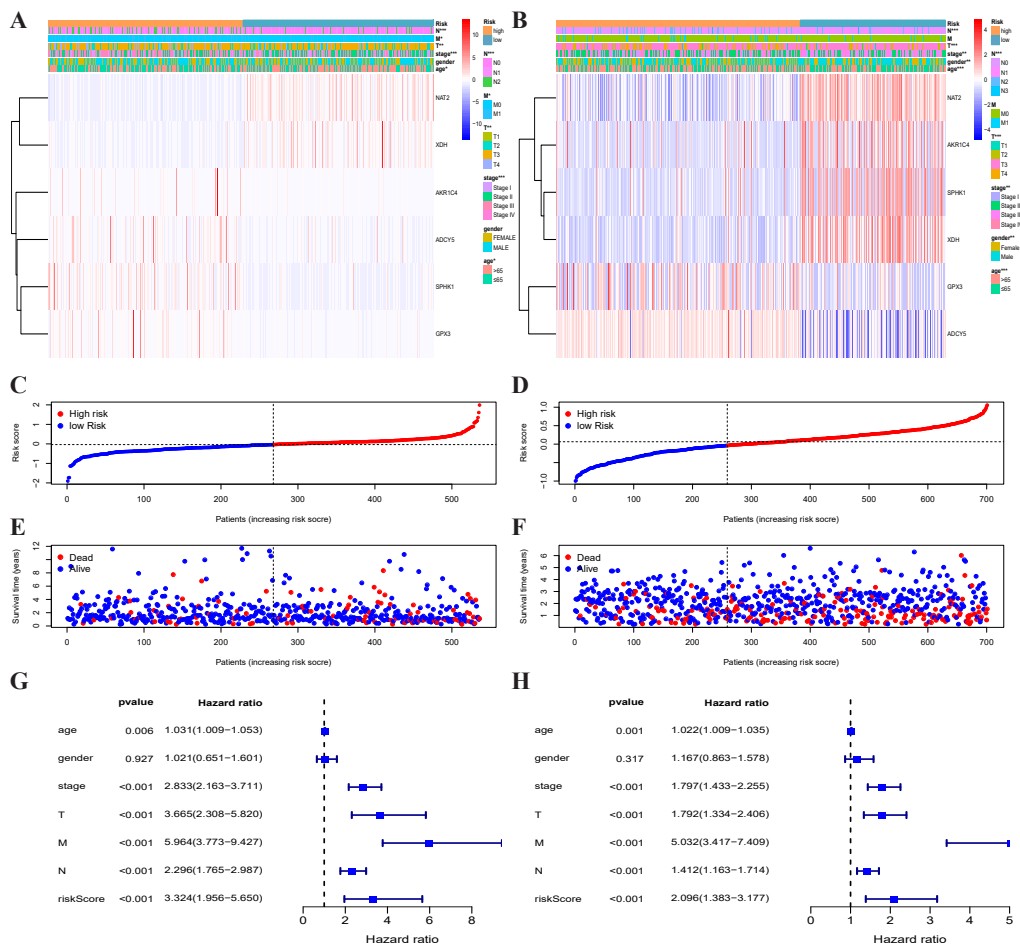

**Figure 3** **Traits of the prognostic metabolism-associated genes signature.** Heatmap of the metabolism-associated gene expression profiles in prognostic signature for TCGA-CRC (A) and GEO-CRC (B). The distribution of risk score and patient's survival time of TCGA-CRC (C) and GEO-CRC (D). The black dotted line is the optimum cutoff dividing patients into low-risk and high-risk groups. The red curve represents high risk and the blue curve represents low risk. The distribution of survival status of TCGA-CRC (E) and GEO-CRC (F). The dots indicate the survival status, the red dot indicates the death of the patient and the blue dot indicates alive. (G, H) Univariate Cox regression analysis. Forest plot of the association between risk factors and survival of TCGA-CRC (G) and GEO-CRC (H). TCGA, the Cancer Genome Atlas database; GEO, Gene Expression Omnibus; CRC, colorectal cancer.

## The risk score and metabolic genes related to the clinicopathological features of CRC

The expressions of NAT2, ADCY5, SPHK1, GPX3, and the risk score were significantly correlated with the clinicopathological features of TCGA-CRC (Table S2, Figs. 5A–5K). NAT2 was highly expressed in stages I, II, $N_0$, and $M_0$, but had low expression in stages III, IV, $N_{1–2}$, and $M_1$ (Figs. 5A–5C) ($p < 0.05$). However, the expression level of ADCY5 was higher in stages III, IV, and $N_{1–2}$ than in stages I, II, and $N_0$ (Figs. 5D and 5E). SPHK1 and GPX3 were also highly expressed in $T_{3–4}$ but had low expressions in $T_{1–2}$ (Figs. 5F and 5G). The risk score was significantly correlated with the clinical stage, which was highly expressed

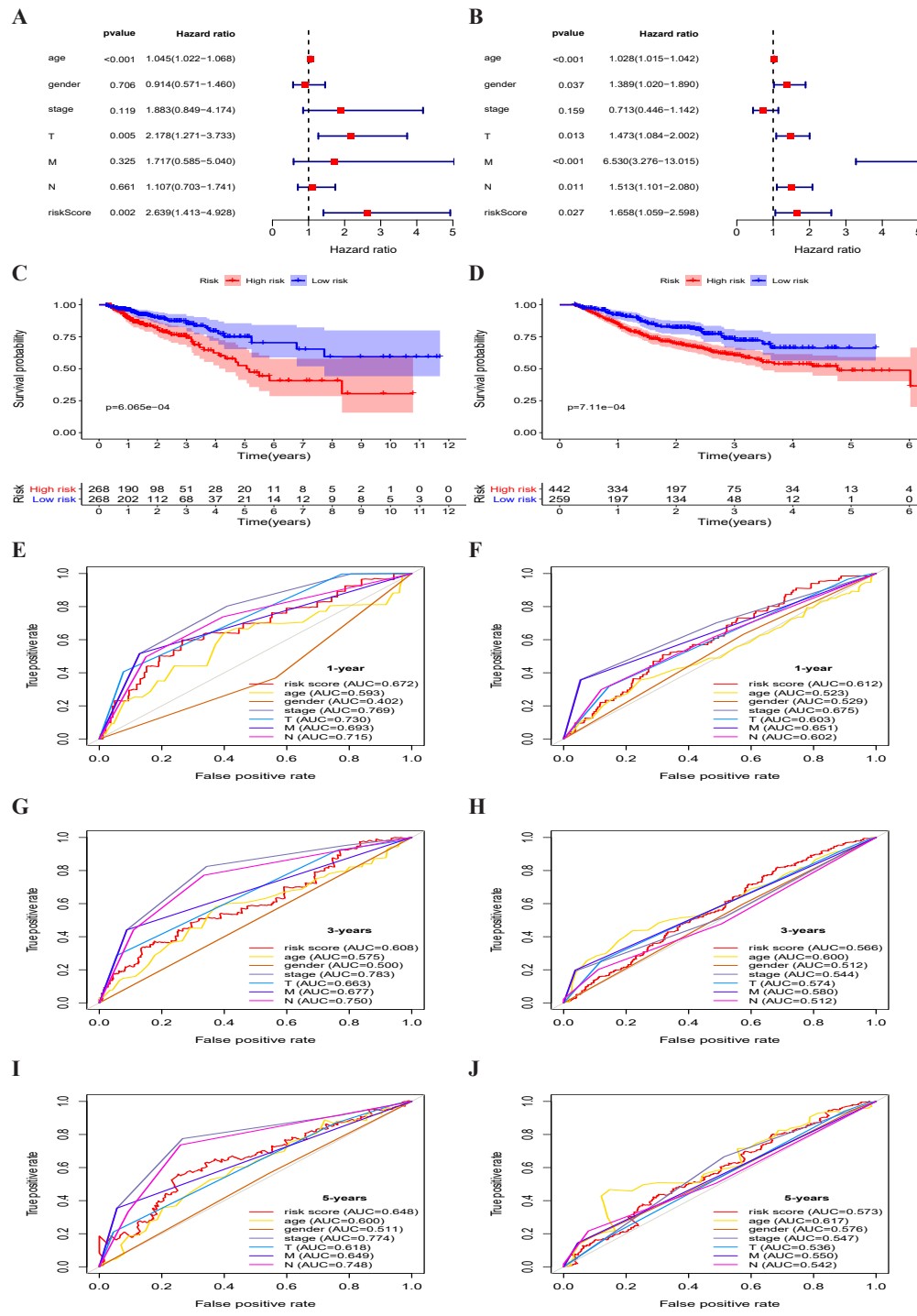

**Figure 4  Metabolism-associated gene signature was significantly associated with survival in colorectal cancer.** (A, B) Multivariate Cox regression analysis. The risk score was an independent prognostic element in TCGA-CRC (A) and GEO-CRC (B). (C, D) Kaplan-Meier survival analysis of 

**Figure 4 (…continued)**
CRC patients ranked by the median risk score. The $X$ axis represents the survival time (year) of the CRC patient; the $Y$ axis represents the survival probability of the CRC patient. The high-risk score was related to poor OS in TCGA-CRC (C) and GEO-CRC (D). ROC analysis of the sensitivity and specificity of the OS for the combination of risk score and clinical characteristics in TCGA-CRC (E, G, I) and GEO-CRC (F, H, J). TCGA, the Cancer Genome Atlas database; GEO, Gene Expression Omnibus; CRC, colorectal cancer; ROC, receiver operating characteristic; OS, overall survival; AUC, area under curve; T, primary tumor; M, distant metastasis; N, regional lymph nodes.

in stages III, IV, $T_{3-4}$, $N_{1-2}$, and $M_1$, but had low expression in stages I, II, $T_{1-2}$, $N_0$, and $M_0$, (Figs. 5H–5K) ($p < 0.05$). These results showed that NAT2 was a low-risk metabolic gene, and ADCY5, SPHK1, GPX3, were high-risk metabolic genes, which is consistent with our previous results. More importantly, the risk score closely correlated with the malignant clinicopathological characteristics of CRC and is an independent prognostic factor.

Similar results were seen in the validation group (Table S3, Figs. 5L–5AA). NAT2 and XDH were highly expressed in the <65-year age group, $M_0$, and had low expression in the >65-year age group, $M_1$ (Figs. 5L–5O) ($p < 0.05$). The expression level of ADCY5 was the opposite of NAT2 and XDH (Figs. 5P and 5Q). SPHK1 and GPX3 were highly expressed in stage III, IV, $T_{3-4}$, and $N_{1-3}$, and had low expressions in stages I, II, $T_{1-2}$, and $N_0$ (Figs. 5R–5W). The expression of AKR1C4 was higher in stages III, IV, and $N_{1-3}$, than in stages I, II, and $N_0$ (Figs. 5X–5Y). The risk score was higher in the group of >65 years old and $M_1$, lower in the group of <65 years old and $M_0$ (Figs. 5Z and 5AA) ($p < 0.05$).

## Identification involved signaling pathways of PRMGs

GO enrichment analysis of the seven metabolic genes showed that there were two GO terms in BP, one GO term in MF, and one GO term in CC, which was significant (FDR < 0.05). GO enrichment analysis showed that these seven genes could be classified into several essential processes, including oxidation–reduction, the xenobiotic metabolic process, electron carrier activity, and cytosol (Fig. 6A). KEGG pathway enrichment shown that PRMGs were mainly enriched in caffeine metabolism, metabolic pathways, and drug metabolism by other enzymes (Fig. 6B) ($p < 0.05$). GSEA analysis showed that changed genes were observably enriched in several common pathways. 98 of 178 gene sets were upregulated in the high-risk phenotype group, and 95 gene sets were remarkable at FDR < 25%. The top-five gene sets of the high-risk group were significantly related to basal cell carcinoma (NES = 2.31, $P = 0.000$), dilated cardiomyopathy (NES = 2.28, $P = 0.000$), vascular smooth muscle contraction (NES = 2.25, $P = 0.000$), glycosaminoglycan biosynthesis chondroitin sulfate (NES = 2.24, $P = 0.000$), and axon guidance (NES = 2.23, $P = 0.000$). 80 of 178 gene sets were upregulated in the low-risk phenotype group. 74 gene sets were markedly enriched at FDR < 25%, and the five most common gene sets of the low-risk group were negatively associated with peroxisome (NES = $-2.37$, $P = 0.000$), fatty acid metabolism (NES = $-2.36$, $P = 0.000$), butanoate metabolism (NES = $-2.31$, $P = 0.000$), valine, leucine, and isoleucine degradation (NES = $-2.26$, $P = 0.000$), and propanoate metabolism (NES = $-2.26$, $P = 0.000$) (Fig. 6C). The integration of the 5 most prevalent phenotypes of the high-risk and the low-risk groups was visualized in Fig. 6D.

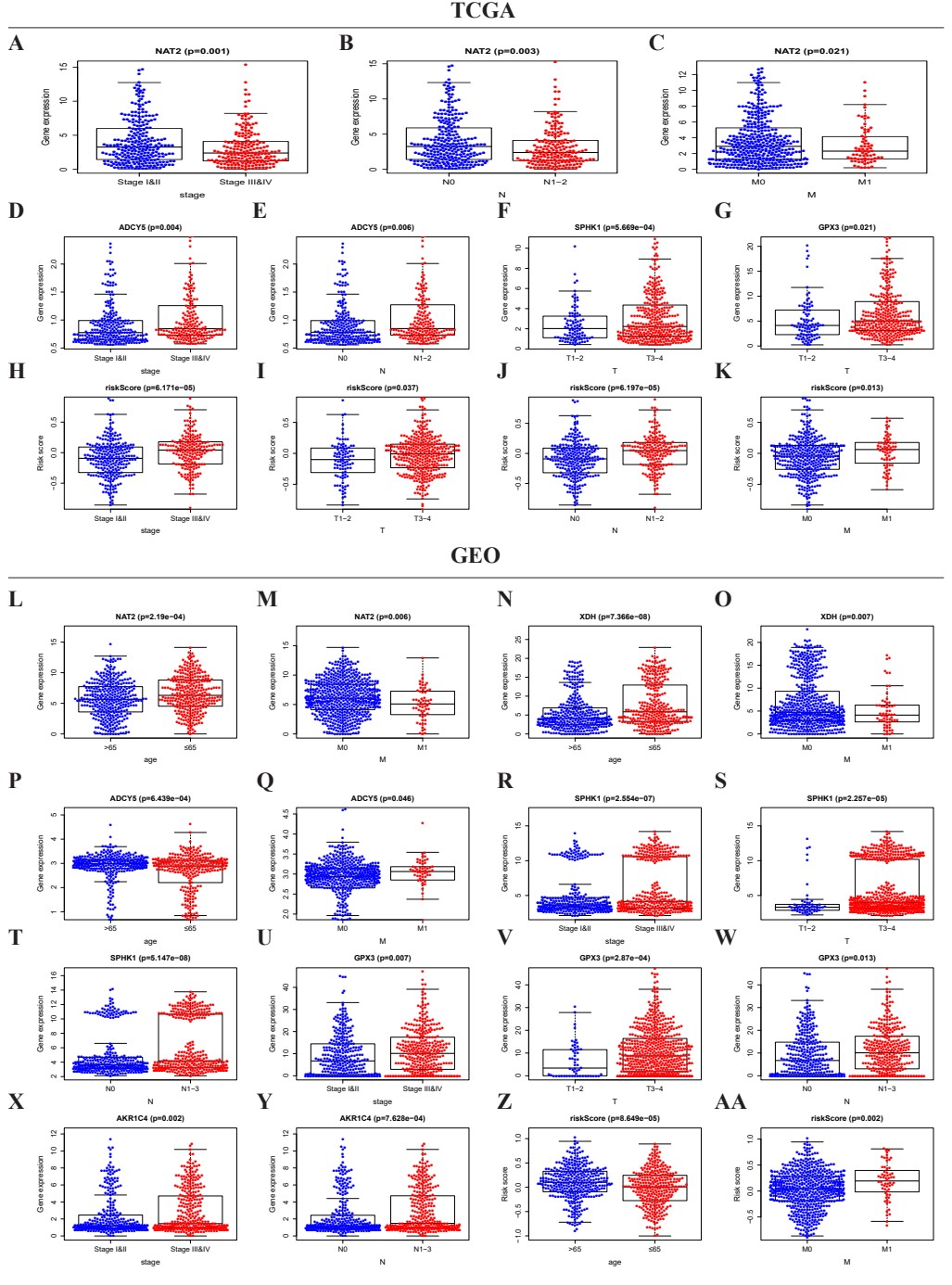

**Figure 5  Risk score and metaboli sm-genes associated with the clinicopathological features of CRC.**
Box-plot showed that there was a significant association between metabolic genes expression, risk score, and clinicopathological features in the TCGA dataset (A–K) and GEO dataset (L–AA). In the TCGA dataset, the expression of NAT2 related to the group of the stage (continued on next page...)

**Figure 5 (…continued)**
(A), N-stage (B), and M-stage (C). The expression of ADCY5 related to stage (D) and N-stage (E); SPHK1 (F), GPX3 (G) associated with T-stage; Risk score associated with stage (H), T-stage (I), N-stage (J), and M-stage (K). In the GEO dataset, the expression of NAT2 related to a group of age (l) and M-stage (M). The expression of XDH related to a group of age (N) and M-stage (O). The expression of ADCY5 associated with age (P) and M-stage (Q); SPHK1 associated with stage (R), T-stage (S), and N-stage (T); GPX3 associated with stage (U), T-stage (V), and N-stage (W); AKR1C4 had a correlation with stage (X) and N-stage (Y). Risk score associated with age (Z) and M-stage (AA).

## A characterized prognostic prediction model

A nomogram is a powerful tool quantifying an individual's risk in a clinical setting by integrating multiple risk factors (*Liang et al., 2015*; *Won et al., 2015*). We used the nomogram to predict the probabilities of 1-, 3- and 5-year OS by incorporating the age, gender, TNM stage, and risk score of TCGA-CRC (Fig. 7A). The results were verified in the GEO-CRC. Each factor was assigned a score in proportion to its contribution to the risk of survival. The calibration curve showed that the actual survival time is in agreement with the predicted survival time and the C-index is 0.8. (Figs. 7B, 7D and 7F). A 75-year old (60 points) female patients (6 points) would acquire a total of 224 points if she had stage III (41points), T3 (32 points), N1 (0 points), and M0 (0 points), with a risk score (85 points). Her 1-, 3-, and 5-year survival rate was approximately 67%, 45%, and 28%, respectively. The nomogram in the GEO colorectal cancer cohorts and the 1-, 3- and 5- year calibration curves are shown in Figs. 7C, 7E and 7G, respectively.

## DISCUSSION

Specific metabolic activities can directly influence the transformational process or support the biological processes that make tumors grow (*Vander Heiden & DeBerardinis, 2017*). Recent data have shown that microbial metabolites, such as secondary bile acids, promote carcinogenesis; metabolic links between gut microbes are associated with cancer and a diet rich in fat and meat; and extracellular metabolic energetics can promote cancer progression, especially in colorectal cancer (*Jia, Xie & Jia, 2018*; *Louis, Hold & Flint, 2014*; *Wirbel et al., 2019*; *Wong & Yu, 2019*). Prognostic predictions are crucial to the selection of clinical treatment regimens for cancer patients. Several studies have explored prognostic biomarkers and found that gene expression profiles play a vital role in the prognosis of cancer (*Jiang et al., 2019*; *Shen et al., 2019a*; *Zhou et al., 2018*).

We attempted to construct a prognostic model for CRC patients based on six PRMGs filtered out by LASSO COX regression analysis and determined the risk score. The prognostic model was accurate and gave a precise predictive value. Our results showed that the risk score is an independent prognostic factor. The expression of NAT2, ADCY5, SPHK1, GPX3, and risk score also was significantly associated with the clinicopathological features of CRC. The prognostic metabolism-associated gene signature was validated in an independent GEO colorectal cancer cohort. We also explored the mechanism of the seven PRMGs through GO term, KEGG pathway enrichment, and GSEA analysis. The results showed that these genes correlated with some metabolic processes and metabolic pathways. Moreover, we constructed a nomogram to predict 1-, 3- and 5-year OS probabilities by

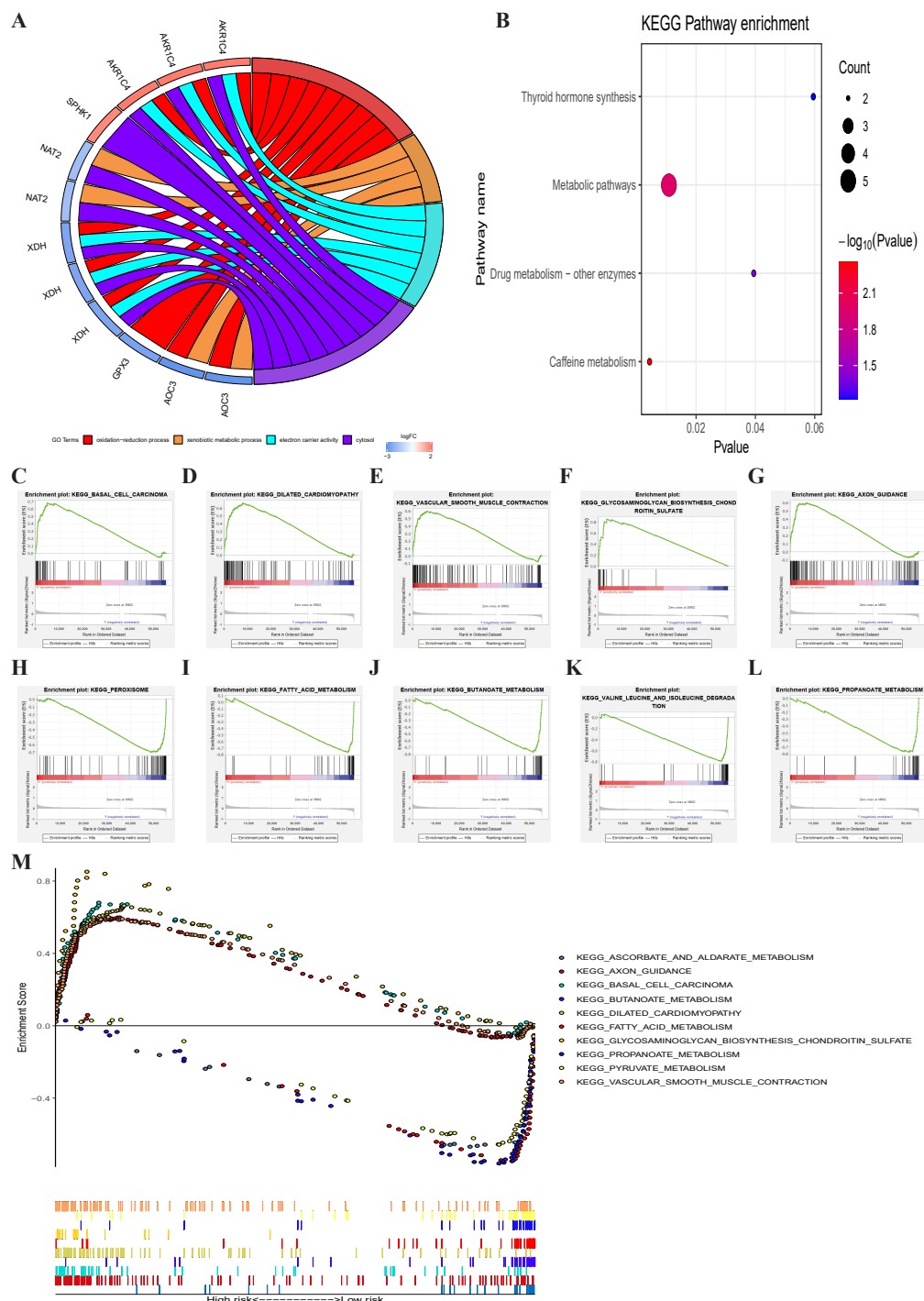

**Figure 6   GO, KEGG, and GSEA analysis.** (A) GO analysis of six PRMGs, on the left of circle chart is the gene, up-regulation was red, and down-regulation was blue, on the right of circle chart is different GO terms, and the genes linked via ribbons to their assigned terms ($p < 0.05$). (B) KEGG pathway of PRMGs. (C) GSEA analysis of the differentially expressed genes between high and (continued on next page...)

**Figure 6 (…continued)**

low-risk groups. Green line chart representation enrichment profile, horizontal axis is each gene under the KEGG pathway, and the vertical axis is the corresponding accumulated enrichment score. The peak in the line graph is the enrichment score of this pathway, and the gene before the peak is the core gene of the pathway; Hits representation mark the genes of the pathway with black lines; Ranking metric scores indicates the distribution of rank values of all genes in the pathway. (D) Multiple GSEA analyses of the differentially expressed genes between high and low-risk groups. GO, gene ontology, KEGG, kyoto encyclopedia of genes and genomes, GSEA, gene set enrichment analysis. PRMGs, prognosis-related metabolic genes.

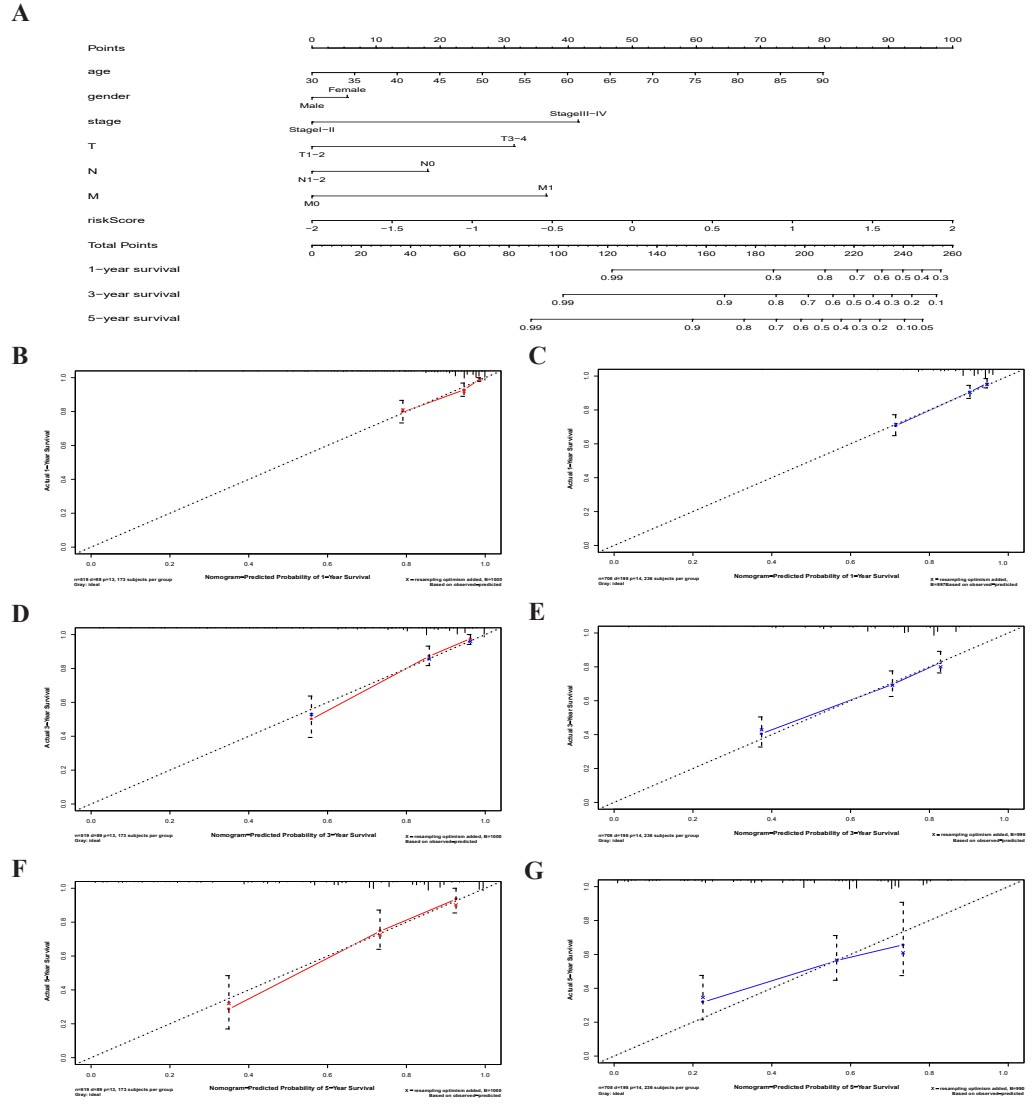

**Figure 7 The nomogram to anticipate prognostic probabilities in CRC.** (A) The nomogram for predicting 1-, 3- and 5-year OS of CRC by clinical-pathological features and Risk score. The 1-, 3- and 5- year calibration curves of TCGA-CRC (B, D, F) and GEO-CRC (C, E, G). *X* axis represents predicted survival time and *Y* axis indicates actual survival time. TCGA, the Cancer Genome Atlas database; GEO, Gene Expression Omnibus; CRC, colorectal cancer. OS, overall survival.

integrating six-metabolic gene signatures and clinicopathological features and the results were validated within the independent cohort of GEO databases.

Univariate Cox regression analysis demonstrated that seven metabolism-related genes were significantly related to the OS of CRC, including five high-risk PRMGs and two low-risk PRMGs. Previous reports have shown that Glutathione peroxidase 3 (GPX3) methylation may play a crucial role in predicting the platinum sensitivity of CRC (*Pelosof et al., 2017*). SPHK1 has an essential role in the development of multifunctional NF-$\kappa$B-regulated cytokine IL-6 and is continuously activated by the transcription factor STAT3. It also plays a role in colitis-associated cancer (*Liang et al., 2013*). There are additional reports that AKR1C4 and ADCY5 are hub genes that may be independent prognosis biomarkers and therapeutic targets for CRC patients (*Gylfe et al., 2013*; *Yang, Zhang & Yang, 2019*), which was consistent with our results.

Whether the expression level of metabolism-related genes can be used as a prognostic maker is a vital topic of research. Our colorectal cancer prognostic model based on six metabolism-related genes was found to be of value and the 1-, 3-, and 5-year risk scores and the AUC values of ROC were consistent with previous reports (*Jeun et al., 2019*; *Park et al., 2009*; *Peng et al., 2019*; *Sun et al., 2019*). Prognoses were classified as high- and low-risk and the six metabolic genes we screened may be ideal prognostic markers. 5-year OS was about 50% in the high-risk group and 75% in the low-risk group, which was consistent with previous reports (*Dueland et al., 2018*). High-risk genes in the model, including GPX3, AKR1C4, SPHK1and ADCY5, have been reported to promote hypermethylation, rare mutations, cancer progression, poor prognosis, and the developmental process of malignant cells in CRC patient samples or cell lines (*Gylfe et al., 2013*; *Kawamori et al., 2009*; *Liang et al., 2013*; *Pan et al., 2019*; *Zhou et al., 2019*). SPHK1 promotes the phosphorylation and activation of p65, thus promoting the occurrence of CRC (*Shen et al., 2019b*). The expressions of NAT2, ADCY5, SPHK1, GPX3, and risk score were significantly correlated with the clinicopathological features of CRC, and the risk score was closely correlated with the malignant clinicopathological characteristics of CRC and is an independent prognostic factor. NAT2 is associated with a high risk of colorectal cancer, mainly due to its involvement in the metabolism of aromatic and heterocyclic aromatic amines in cooked red meat (*Lilla et al., 2006*). We also identified six metabolically-related genes that were significantly correlated with gene expression and prognosis in CRC patients in the GEO database (a separate cohort of 720 CRC patients). GPX3, AKR1C4, and SPHK1 are reported to have involvement in the pathogenesis of CRC and in predicting overall survival, reinforcing the prognostic value of our TCGA and GEO cohort analysis. The remaining ADCY5 gene has not been associated with CRC prognosis and may be used as a potential biomarker for CRC.

Our study identified the metabolic genes associated with the GO and signaling pathways of CRC. Several crucial processes and signaling pathways have been identified by GO enrichment analysis, caffeine metabolism, metabolic pathways, and KEGG pathway analysis including oxidation–reduction and the xenobiotic metabolic process. Previous research demonstrated that the oxidation–reduction process and xenobiotic metabolic process play

a crucial role in the development of CRC or colorectal cancer cells (*Bensard et al., 2020*; *Han et al., 2016*).

We constructed a nomogram to predict individual clinical outcomes. The nomogram is a stable and reliable tool for quantifying individual risk by combining and describing risk factors. It has been used for tumor prognosis, including for CRC (*Renfro et al., 2017*; *Sjoquist et al., 2018*). The nomogram generates a graphical statistical prediction model that assigns scores to each factor, including age, sex, and clinical stage. The model summarizes all clinical points to provide numerical possibilities for clinical outcomes such as OS, relapse, and drug nonadherence. In addition to traditional clinicopathologic features such as TNM staging, tumor size, and histological subtypes, risk scores based on genetic markers can also be incorporated into a predictive nomogram model to predict clinical outcomes (*Reichling et al., 2019*; *Sjoquist et al., 2018*). A nomogram predicted 3- and 5-year recurrence-free survival rates for non-small cell lung cancer and gave a prognostic score calculated by the autophagy gene signature (*Liu et al., 2019*). The combination of autophagy gene characteristics and prognostic factors has a better prognostic value than the single application (*Mo et al., 2019*). Calibration curves showed that the nomogram, which included RNA signals and conventional prognostic factors, accurately predicted 3- and 5-year survival probabilities (*Xiong et al., 2017*). Our nomogram, which includes risk scores and clinicopathologic features, is a good predictor of survival in 1-, 3- and 5-year CRC patients.

We constructed a six metabolic gene model for colorectal cancer patients based on TCGA to predict the prognosis of colorectal cancer patients. A risk score based on six genes may be a promising independent prognostic biomarker and is closely correlated with the malignant clinicopathological characteristics for CRC patients. A nomogram based on genetic characteristics and clinicopathologic characteristics may accurately predict the survival probability of individual CRC patients at 1, 3, and 5 years. Some of the programmatic improvements to the model can be made at other levels to represent scheduling activities in more detail. Further research on these genes may improve their clinical application and may provide a new perspective into the pathogenesis of CRC. Some of the antecedently overlooked genes may be additional biomarkers for CRC and require further study. Our study improved our understanding of the interactions between CRC cells and the tumor metabolism microenvironment and may identify novel therapeutic targets.

## CONCLUSION

We comprehensively identified PRMGs, constructed a six-metabolic gene model, and analyzed their molecular function in CRC. Our study also highlighted the crucial role of the risk score as an independent prognostic biomarker that is closely correlated with the malignant clinicopathological characteristics for CRC patients. Our research identified several crucial processes and signaling pathways of the metabolic genes in CRC. These findings provide a comprehensive outlook for further studies into the roles of metabolic genes in the pathogenesis of CRC and as potential biomarkers for CRC diagnosis and therapeutics.

## ACKNOWLEDGEMENTS

We thank the TCGA and GEO databases for the availability of the data.

### Funding

The authors received no funding for this work.

### Competing Interests

The authors declare there are no competing interests.

### Author Contributions

- Yandong Miao conceived and designed the experiments, analyzed the data, prepared figures and/or tables, authored or reviewed drafts of the paper, and approved the final draft.
- Qiutian Li conceived and designed the experiments, prepared figures and/or tables, and approved the final draft.
- Jiangtao Wang and Yuan Yang performed the experiments, prepared figures and/or tables, and approved the final draft.
- Wuxia Quan and Chen Li analyzed the data, prepared figures and/or tables, and approved the final draft.
- Denghai Mi conceived and designed the experiments, authored or reviewed drafts of the paper, and approved the final draft.

### Data Availability

The data are available at NCBI GEO (GSE39582, GSE87211) and at TCGA.

The search instructions for TCGA:

(1) Case keywords: "colon, rectosigmoid junction, and rectum [Primary Site]", "TCGA[Program]", "TCGA-COAD, TCGA-READ [Project]", "Adenomas and Adenocarcinomas [Disease Type]".

(2) File keywords: "Transcriptome Profiling [Data Category]", "Gene Expression Quantification [Data Category]", "RNA-Seq [Experimental Strategy]", "HTSeq—FPKM [Workflow Type]", "Clinical [Data Category]", "BCR XML [Data Format]".

### Supplemental Information

Supplemental information for this article can be found online at http://dx.doi.org/10.7717/peerj.9847#supplemental-information.

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
