# Peer review of "Prognostic implications of metabolism-associated gene signatures in colorectal cancer"

_PeerJ, doi:10.7717/peerj.9847_

## Round 0.1 · original submission · Major Revisions

I find this to be a well-written manuscript that should be suitable for publication once all of the reviewers' comments have been adequately addressed.

Reviewer 1 ·

Basic reporting

The paper by Miao et al described the identification of prognosis-related metabolic genes for colorectal cancer using available TCGA and GEO datasets and evaluated their potential roles in disease pathogenesis. The manuscript is written well and covered all aspects of the area that need to be reviewed before publication. I have a minor comment which is described below and overall, I recommend this article for publication.

The resolution of figures 1, 2, 3, 5 and 6 are very poor. It is very hard for this reviewer to read the contents of figures. Please expand all the figure legends and appropriately describe X and Y axis in the figures.

Experimental design

No comment.

Validity of the findings

No comment.

Reviewer 2 ·

Basic reporting

- There still has some grammatical errors and typos. The authors should re-check and revise carefully.
- Quality of figures needs to be improved.
- Background and literature review are weak, need to add more information.
- The authors should provide source codes for reproducing the results. It is very important.

Experimental design

- The flowchart (Fig. 1) looks unclear. For example, did the clinicopathology data come directly to DEG analysis? As I known, it is maybe only gene data.
- ROC curve or AUC has been used in previously biomedical works such as PMID: 31277574 and PMID: 31362508. Therefore, it is suggested to add more references in this description.

Validity of the findings

- It is easy to see that the performance results are not quite significant (i.e., very low AUC). Therefore, it's hard to convince that this gene set is significant.
- Also, ROC curves showed that the model is not so good.
-

Additional comments

No comment.

Reviewer 3 ·

Basic reporting

NA

Experimental design

NA

Validity of the findings

NA

Additional comments

In this manuscript “Prognostic implications of metabolism-associated gene signatures in colorectal cancer” Miao et al. have identified a prognostic model in colorectal cancer patients based on metabolic genes. They showed that risk score that authors have identified based on TCGA is robust across GEO datasets and correlated with clinic pathological characteristics. Authors have performed multiple bioinformatics analysis to characterize the risk model but completely lack rationale and key methodological details which makes manuscript hard to follow in its current form. Following are the points to be addressed:

Major comment:

1. Abstract of the manuscript is verbose and lacks specificity. Abstract should be written in a clear and consise way, explaining the research question and results/conclusions.

2. Authors have not mentioned how prognosis associated metabolism related genes were identified? Was comparison pf CRC patients was done with normal? Line 86 -> Define what is data extraction and integration here? What pathways are enriched in Differentially expressed PRMGs? Does these metabolism associated genes are also differentially expressed in different subtypes of CRC?

3. What is the significance of using Gene Set Cancer Analysis 
when mutation patterns in genes can be easily mapped using TCGA mutation information of CRCs? A simple bargraph representing the mutation frequency can also show the importance of picked genes. Cite appropriate references to discuss the importance of selected genes and there known role in CRC if any.

4. Line 65-> which genes? Explain why there is a need of a new prognostic model? Define PRMGs in introduction section of the manuscript. What is rationale of using metabolism related genes?

5. What is advantage of using risk score over patients clinical characteristics when five year survival statistics is similar with either (Figure 3)?


Minor comments:

6. What level of TCGA was used for the analysis? Does authors perform expression normalization?

7. Table1: Mainly pathway -> Pathway

8 Figure quality is very poor and text is unreadable.

9. Figure1: Figure quality is very poor. 1E what does X and Y axis indicates?

10. How patients were divided into low-high risk groups? Are these six prognostic genes are known prognostic genes for other tumor types?


11. Why flowchart of study design is Figure 7 of the paper? It should be the first figure explaining the schematic of study design.

---

## Round 0.2 · accepted · Accept

Thank you for thoroughly addressing the reviewers' comments. They have made no further suggestions and both recommend to accept as is. Well done.

Reviewer 2 ·

Basic reporting

No comment

Experimental design

No comment

Validity of the findings

No comment

Additional comments

My previous comments have been addressed satisfactorily.

Reviewer 3 ·

Basic reporting

NA

Experimental design

NA

Validity of the findings

NA

Additional comments

Authors have addressed all my concerns